# Thyroid Hormones and Health-Related Quality of Life in Normal Pressure Hydrocephalus Patients before and after the Ventriculoperitoneal Shunt Surgery: A Longitudinal Study

**DOI:** 10.3390/jcm11154438

**Published:** 2022-07-30

**Authors:** Mindaugas Urbonas, Nijole Raskauskiene, Vytenis Pranas Deltuva, Adomas Bunevicius

**Affiliations:** Neuroscience Institute, Lithuanian University of Health Sciences, 44307 Kaunas, Lithuania; nijole.raskauskiene@lsmuni.lt (N.R.); vytenis.deltuva@lsmuni.lt (V.P.D.); adomas.bunevicius@lsmuni.lt (A.B.)

**Keywords:** low T3 syndrome, hydrocephalus, SF-36, thyroid hormones, iNPH, ventriculoperitoneal shunt surgery

## Abstract

Objective: The aim of this study was to explore the serum levels of thyroid-stimulating hormone (TSH), free thyroxine (FT4), and free triiodothyronine (FT3), and to correlate the hormone levels among iNPH patients with their self-reported quality of life before and three months after the surgery. Methods: Twenty-five patients (52% women), mean age 63.5 (SD 9.5) years, were operated on by inserting a VP shunt. Patients with FT3 level ≤3.34 pmol/L were diagnosed as having low T3 syndrome. Results: The changes in thyroid hormones resulted in a U-shaped curve throughout the follow-up period. The significant changes occurred the next day after the surgery, including a decrease in TSH, FT3, and an increase in FT4. Additionally, the decrease occurred in mean FT3 for six patients with preoperative low T3 syndrome. Three months after the surgery, thyroid hormones were restored to their baseline and/or normal values. All six patients with preoperative low T3 syndrome had significant improvement in all SF-36 subscales (except for the role emotional and physical). Patients with preoperative normal high FT3 and low FT4 had increased FT3/FT4 ratio which was associated with deterioration in all SF-36 subscales 3 months after the surgery. Conclusion: Routine assessment of the FT3/FT4 ratio might be a simple and effective tool for the risk stratification of iNPH patients before VP shunt surgery.

## 1. Introduction

Normal pressure hydrocephalus (NPH) is a brain disorder in which excess cerebrospinal fluid (CSF) accumulates in the brain’s ventricles and it is characterized by the phenotypic triad of progressive gait disturbance, cognitive impairment, and urinary incontinence associated with normal CSF pressure and ventricular system dilation that cannot be attributed to cerebral atrophy, although episodes of increased CSF pressure do occur [1,2].

Impaired perfusion in the periventricular tissue, reduced metabolism of glucose, declined level of N-acetylaspartate in the basal ganglia, neuroinflammation, and impaired function of the glymphatic system have recently been suggested as possible pathophysiological mechanisms in iNPH [3,4,5,6]. Third ventricle adjacent structures and connected structures, such as the anterior cingulate cortex, are considered to be associated with the symptoms and signs [7,8,9,10].

Hypothalamo-hypophyseal dysfunction in iNPH patients was reported by Barber and Garvan [11]. Five patients out of six in the study of Barber and Garvan showed endocrine disturbance, mainly hypogonadism. Moin et al. found pituitary dysfunction in 31% of iNPH patients, most commonly hypogonadism [12]. In addition, Ucler et al. revealed that hydrocephalic newborns had higher levels of TSH and FT3 when compared with the healthy control group before and after the ventriculoperitoneal (VP) shunt surgery [13]. The levels of TSH became lower in the shunting group 3 months after the operation when compared with the control group. Ucler et al. considered that biochemical and anatomical changes occurring in hydrocephalus could affect the hypothalamic–pituitary–thyroid axis and insertion of a VP shunt would facilitate a better physiological environment for the hypothalamic–pituitary axis.

Low T3 syndrome is a common complication in patients with intracranial neurosurgical disorders. It has been well established that in patients suffering an acute stroke, low T3 serum concentration was associated with unfavorable functional outcomes by discharge or during follow-up [14]. Recent investigations showed that reduced T3 and TSH concentrations are associated with impaired quality of life in brain tumor patients [15]. Uncertainty remains about whether nonthyroidal illness syndrome is an adaptive state that promotes recovery or a genuine organ failure indicating maladaptation of the organism to surgical stress [16]. However, there is a paucity of data in the literature regarding the function of the hypothalamic-pituitary-thyroid axis and health-related quality of life in normal pressure hydrocephalus patients.

The aim of this study was to explore the serum levels of thyroid-stimulating hormone (TSH), free thyroxine (FT4), and free triiodothyronine (FT3) and to correlate the hormone levels among iNPH patients with their self-reported health-related quality of life before and three months after the surgery.

We hypothesized (1) that patients with iNPH would display an early postoperative adverse response in the studied hormones including a decrease in FT3, TSH, and an increase in FT4 levels with respect to the baseline; (2) that changes in thyroid hormones before and during the follow-up after the VP shunt surgery would have an influence on health-related quality of life.

## 2. Materials and Methods

### 2.1. Design and Procedure

All the patients who required treatment for NPH at the Department of Neurosurgery of Hospital of Lithuanian University of Health Sciences, Kaunas, Lithuania, in the period from January 2019 to March 2021 were recruited. The patients were originally diagnosed in a way that closely corresponded to the iNPH guidelines criteria for probable iNPH [17]. The diagnosis was based on a clinical presentation (clinical triad: typical gait disturbance, cognitive impairment, urinary incontinence), neuroimaging, and CSF investigation. The degree and severity of clinical symptoms were evaluated by the iNPH grading scale (iNPHGS) [18]. Gait disturbance was evaluated using a timed up-and-go test (TUG) and a short-distance straight walking test [19]. The Mini-Mental State Examination (MMSE) was used for the evaluation of cognitive impairment [20]. MRI showed communicating ventriculomegaly (Evans index > 0.3) as well as temporal horn enlargement, periventricular signal changes, periventricular edema, or an aqueductal/fourth ventricular flow void. The lumbar CSF opening pressure was below 245 mm H_2_O and CSF examination was normal. All the patients underwent the spinal tap test (40–50 mL of CSF removed) with the evaluation of clinical symptoms using the TUG test, short-distance walking test, and MMSE. The exclusion criteria were as follows: a history of thyroid disease; history of preexisting pituitary dysfunction relating to any cause; major neurological or psychiatric conditions that could potentially affect cognition. Patients were operated on for hydrocephalus by inserting a ventriculoperitoneal (VP) shunt with a programmable valve (Miethke proGAV 2.0 (Miethke, Potsdam, Germany), Codman Hakim with Siphonguard).

Patients underwent head MRI before the surgery, 4–5 days after the surgery, and 3 months after the ventriculoperitoneal shunt surgery. The readings greater than 0.3 for the Evans index (EI) define ventricular enlargement and have been used as a measure of ventriculomegaly. The EI is defined as the ratio of the maximal width of the frontal horns to the maximum inner skull diameter on the same plane on MRI or CT.

### 2.2. Self-Reported Health-Related Quality of Life

All patients within 3 days before and 3 months after the surgery were evaluated for quality of life with the SF-36 questionnaire. Patients were asked to complete the SF-36 questionnaire themselves and were given an opportunity to ask questions if any. The SF-36 questionnaire is a generic health status profile measure consisting of eight subscales: general health (GH); bodily pain (BP); physical functioning (PF); role physical (RP), mental health (MH); vitality (V); social functioning (SF); role emotional (RE). Each of the eight SF-36 subscales ranges from 0 to 100, with higher scores indicating better health-related quality of life [21]. A reliability analysis was carried out on the SF-36 scale comprising eight items. Cronbach’s alpha showed the questionnaire reached acceptable reliability, α = 0.89. Adequate construct validity and psychometric properties of the Lithuanian translation of the SF-36 were previously documented in patients with brain tumors [22].

### 2.3. Blood Samples

All blood samples were drawn from the patients within 3 days preoperatively, the next day after the operation, and 3 months after the operation in a fasting state. Free triiodothyronine (FT3), free thyroxine (FT4), and thyroid-stimulating hormone (TSH) were measured in the serum by fluorometric enzyme immunoassay method with a TOSOH AIA 2000 Immunoassay Analyzer (Tosoh Bioscience, Inc., South San Francisco, CA, USA). Reference values provided by the laboratory: FT3 (3.34–5.14 pmol/L), FT4 (9–21.07 pmol/L), TSH (0.4–3.6 mU/L). If patients were found to have an FT3 level of 3.34 pmol/L or less, they were diagnosed as having a low T3 syndrome.

### 2.4. Statistical Analysis

All statistical analysis was performed using the SPSS statistical software (version 17.0, SPSS Inc., Chicago, IL, USA), and a *p*-value of less than 0.05 was considered to be statistically significant. The distribution of measurements was assessed using the Kolmogorov–Smirnov test. Associations between continuous variables were assessed by Pearson product–moment analysis (Pearson r) or Spearman rank correlation analysis (Spearman r), as appropriate. Differences of continuous and categorical variables were analyzed by analysis of variance or Kruskal–Wallis test and χ^2^ test or Fisher’s exact test, as appropriate. Student’s paired *t*-test was used to examine differences between dependent pairs. Non-normal data were evaluated using the Wilcoxon signed-rank test for dependent (i.e., paired) samples. The paired proportion was evaluated using Cochran’s Q test.

To check linearity, associations between FT3 and FT4 were plotted. A scatter plot was used for visualization of the relationship between variables FT3 and FT4. The area-of-scatter plot was divided into four segments (S) according to medians of the variables: S1: LowFT3/LowFT4; S2: LowFT3/HighFT4; S3: HighFT3/HighFT4; S4: HighFT3/LowFT4; where low: below median; high: above median. All patients were stratified by segments for analysis of the effect size of changes in the SF-36 subscale scores before and 3 months after the VP shunt surgery.

The primary outcome was the change in the SF-36 subscales 3 months after the surgery from the baseline condition. Standardized difference effect size and Cohen’s d are expressed in terms of standard deviation units. Cohen’s d of greater than 0.8 were considered large, 0.5 to 0.79 were considered moderate, 0.2 to 0.49 were considered small and Cohen’s d less than 0.2 were considered trivial [23].

Multiple regression analyses were performed to assess which factors affect QoL 3 months after the surgery, using the subscales from SF-36 as dependent variables and age, sex, preoperative FT3, FT4, FT3/FT4 ratio, Evans index, and preoperative scores of the SF-36 subscales as the independent variables. Variance inflation factor (VIF) was evaluated to rule out multicollinearity between the included parameters.

## 3. Results

### 3.1. Characteristics of Patients

Twenty-five patients were included. Women represented 52% (N = 13) of the sample. The mean age of the patients was 63.5 (SD 9.5) years (range 46 to 76 years) and there were no differences in this respect between the male and female patients (59.7 years (range 45 to 71 years) vs. 67.1 years (range 48 to 76 years), respectively) (t = 4.26, *p* = 0.051). The median Evans index was 0.42 (range 0.34 to 0.56).

The results of detailed clinical examinations by a neurologist and a physiotherapist preoperatively and 3 months postoperatively following standardized protocols are shown in Table 1. All the 25 patients met the diagnostic criteria in the International and Japanese Guidelines for iNPH. Improvement was the most pronounced in the gait domain.

### 3.2. Evans Index and Thyroid Hormones Levels

Significant changes in the Evans index (EI) after the VP shunt surgery indicate that patients benefitted clinically from the VP shunt during the follow-up. The EI decreases significantly in the early period (pre- vs. postop, *p* = 0.002 and pre- vs. 3 months, *p* < 0.001) after VP shunt surgery (Table 2).

Thyroid hormone levels on admission, after VP shunt surgery, and at the 3-month follow-up, and their comparisons are shown in Table 2. Admission thyroid hormones were compared to the hormone levels after a 3-month follow-up and no significant differences were found. When the hormones were analyzed separately, we found that FT3 levels were below the reference value (low T3 syndrome) in 24%, 64.5%, and 8% of the patients on admission, after the surgery, and at the 3-month follow-up, respectively (Cochran’s Q test *p* < 0.001). TSH levels were below the normal limits in 4.5% (*n* = 1) of the patients on admission and 18.2% (*n* = 4) of the patients after the surgery. TSH values were found to be increasing in one patient on admission compared to the reference values. Two (9.1%) of patients before the surgery and five (23.8%) of patients after the surgery had a high FT4 level (>21.07 pmol/L) with normal TSH levels.

When comparing preoperative and postoperative thyroid hormone profiles, a significant decrease occurred in the concentration of FT3 (by 0.72 pmol/L) and the concentration of thyroid-stimulating hormone (by 0.54 mIU/L), while a significant increase occurred in the concentration of FT4 (by 2.75 pmol/L) (all *p* < 0.001). After the 3-month follow-up, thyroid hormone levels were restored to the baseline value (Table 2 and Figure 1). Sequential changes in thyroid hormones are shown in Figure 1.

### 3.3. Low T3 Syndrome

The incidence of preoperative low T3 syndrome (FT3 ≤ 3.34 pmol/L) in iNPH patients was 24% (N = 6). The level of preoperative FT4 in these patients was normal, below median (15.18 pmol/L) (Figure 2). Among those with low T3 syndrome, one patient had suppressed TSH (0.24 mIU/L). Moreover, the next day after the surgery the FT3 levels of these patients (2.63 ± 0.27 pmol/L) were found to be significantly lower than preoperative FT3 levels (3.01 ± 0.23 pmol/L; t = 4.98 df = 5 *p* = 0.004). The frequency of low T3 syndrome increased from 24% before shunting to 64% (N = 16) the next day after the surgery (*p* = 0.002).

After the 3-month follow-up, the thyroid hormone FT3 levels returned to baseline value (4.09 ± 0.76 pmol/L) and only one patient had low T3 syndrome (2.47 pmol/L). Five patients out of six with preoperative low T3 syndrome showed a significant increase in FT3 to the normal range (3.01 ± 0.23 vs. 3.43 ± 0.5 pmol/L; t = −2.79, df = 5, *p* = 0.038) (Table 1 and Figure 1).

### 3.4. Relationship between Preoperative FT3 and FT4

The median preoperative level of FT3 in our patients was 3.68 (range 3.24–4.47 pmol/L), and of FT4 was 15.48 (range 14.47–17.97 pmol/L). Patients in the lowest segment (S1, LowFT3/LowFT4) were younger (not significantly) compared with the higher segments (on average, 60.4, 66.5, 61.5, and 66.0 years, for the S1, S2, S3, and S4 segment, respectively; *p* = 0.617) (Figure 2).

### 3.5. Correlation of the Thyroid Hormones with Evans Index

Preoperative Evans index was related to FT3 (r = 0.504, *p* = 0.017), but was not correlated with FT4 (r = 0.015, *p* = 0.949) and TSH (r = 0.261, *p* = 0.241). The bar graph shows (Figure 3) the preoperative Evans index according to the segments of the relationship between FT3 and FT4 levels in the study population. The differences between the segments were not statistically significant.

The bar graph (Figure 4) shows the mean FT3/FT4 ratio before and 3 months after VP shunt surgery. The results showed significant differences in the preoperative FT3/FT4 ratio between segment S4 (HighFT3/LowFT4) and all other segments (all *p* < 0.01). No significant segment-related differences in the FT3/FT4 ratio were detected 3 months after the surgery (S2 vs. S3 and S4 *p* < 0.1) (Figure 4).

### 3.6. Outcomes with Respect to Health-Related Quality of Life

Overall, the unadjusted preoperative and 3-month follow-up mean scores for all SF-36 subscales are shown in Table 3. For the overall population, SF-36 scores at the 3-month follow-up were significantly higher than the baseline for all the subscales except the RP and RE (all *p* < 0.05), with improvements ranging from 4.9 points for the MH to 13.5 for the SF (Table 2).

On average, after VP shunt surgery, moderate effect sizes (ES ≥ 0.5) were found in the SF-36 subscales measuring the GH, SF, and BP scores. Small to moderate (0.20–0.49) effect sizes were found in subscales measuring the PF, V, and MH. The RP and RE scores remained unchanged during the follow-up (Table 3). Only the effect size of the perception of the GH was significantly linked to the levels of preoperative Evans index (r = −0.489 *p* = 0.015).

### 3.7. Preoperative Low T3 Syndrome and Its Effect on Outcome

Out of 25 patients, 6 with low T3 syndrome had significantly higher scores on the preoperative PF (*p* = 0.018) and SF (*p* = 0.032) subscale in comparison to the patients without low T3. Overall, 3 months after the surgery these patients showed improvement in all of the SF-36 subscales from 7 to 33 points, except in the RP which worsened by −6 points (Figure 5).

### 3.8. Preoperative FT3/FT4 Ratio and Its Effect on Outcome

The effect size was analyzed separately among the segments of the relationship between FT3 and FT4. There was a statistically significant effect size of VP shunt surgery on the PF, SF, and GH in segments S1, S2, and S3.

All analyses showed statistically significant estimated average differences in the direction of improvement following the VP shunt surgery in the presence of moderate to high effect size “Cohen’s d” (Table 4 and Figure 6). Overall, no improvement was noted in the RP subscale (effect size from –0.45 to 0.107).

Three months after the surgery, patients with a high preoperative FT3/FT4 ratio (0.322 ± 0.041 pmol/L) in segment S4 with higher than median FT3 and lower than median FT4 demonstrated worsened QoL compared with preoperative QoL. These patients had a negative effect size (from −0.09 to −0.45) in all SF-36 subscales (Table 4).

Among patients with lower than median FT3 and higher than median FT4 (S2), there was a moderate to large improvement in the GH, PF, SF, RE, V, and BP (effect size from 0.508 to 1.127) (Table 4).

Effect sizes of the changes in SF-36 scores at 3 months after the surgery from the baseline are shown in Table 4. Overall, multiple linear regression analysis showed that a high level of preoperative FT3/FT4 ratio was an independent predictor (β = −0.531, t = −3.52, *p* = 0.002) for the SF improvement, and preoperative low FT3 level (β = −0.458, t = −2.30, *p* = 0.032, R^2^ = 0.17) was associated with the GH improvement at the 3-month follow-up after the VP shunt surgery. All other SF-36 subscales were positively associated with preoperative health status. Two subscales, the MH and V, showed an association with the age at VP shunt surgery so the association between the initial score and the score 3 months after the surgery was the strongest among younger patients (β = −0.229, *p* = 0.026 and β = −0.404, *p* = 0.006, respectively). In addition, the V was associated with female gender (β = 0.269, *p* = 0.043) (Table 5).

When the patients were reanalyzed after excluding six patients with preoperative low T3 syndrome (FT3 ≤ 3.34 pmol/L), a better GH subscale score was stronger than in overall population (N = 25) and inversely associated with FT3 (β = −0.554, t = −2.486, *p* = 0.026, R^2^ = 0.257).

## 4. Discussion

According to the data of our study, the changes in thyroid hormones resulted in a U-shaped curve throughout the follow-up period. The significant changes occurred the next day after the surgery, including a decrease in TSH, FT3, and an increase in FT4. Additionally, the decrease occurred in mean FT3 for six patients with preoperative low T3 syndrome (with respect to the baseline, all *p* < 0.001). Three months after the surgery thyroid hormones are restored to their baseline and/or normal values. All six patients with preoperative low T3 syndrome had significant improvement in all SF-36 subscales (except for the role emotional and physical). Patients with preoperative normal high FT3 and low FT4 had increased FT3/FT4 ratio which was associated with deteriorating in all SF-36 subscales 3 months after the surgery.

To the best of our knowledge, we could not find any data in the literature regarding low T3 syndrome in hydrocephalus patients who had a VP shunt surgery. Yang et al. in a study with iNPH patients reported that preoperative women had lower TSH levels and a trend towards higher FT4, both within the reference range, compared with controls [24]. However, this study did not evaluate the hormone levels after the VP shunt surgery. Ucler and coworkers presented the study with hydrocephalic newborns who underwent a VP shunt surgery and thyroid hormones were investigated before and 30 and 90 days after the surgery [13]. The TSH level was higher preoperatively compared to the control group and this value significantly decreased 3 months after the surgery. According to the Ucler data, there were no significant differences between preoperative and postoperative FT3 and FT4 values in the patient group. Importantly, Moin’s study found that endocrine dysfunction may be reversible in some NPH patients 3 months after the neurosurgical correction [12]. There are only a few studies that reported decreased FT3 in children with obstructive hydrocephalus who underwent endoscopic third ventriculostomy [25,26]. Garg et al. presented that 14% of the patients (*n* = 2) had low T3 syndrome before the operation [25]. Two months after the surgery the hormone profiles were normal. According to the study data, one patient with preoperative low T3 deteriorated during the further 1.5-year follow-up. In the study of Fritsch, 32% (*n* = 6) of patients had low T3 syndrome after the surgery [26]. The mean time interval between the endoscopic third ventriculostomy and endocrine evaluation was 25 months. The hormones were not investigated before the surgery. The outcome was evaluated as failed in two patients out of six with low T3 syndrome. Summarizing the data of previous literature concerning obstructive hydrocephalus and low T3 syndrome, there is a trend seen in complicated clinical courses when a patient is diagnosed a low T3 syndrome.

The results from our longitudinal study showed that FT3 levels were below the reference value in 24%, 64.5%, and 8% of the patients on admission, after the surgery, and at the 3-month follow-up, respectively. One patient had suppressed TSH among those with preoperative low T3. The changes in the TSH and FT3 hormones resulted in a U-shaped curve throughout all the follow-up periods. The same U-shape curve of thyroid function is reported by Al-Sofyani and coworkers in children who underwent cardiac surgery [27]. Thyroid hormones were measured preoperatively and 24, 48, and 72 h after the surgery. According to the study, the most significant changes occurred 24 h after the surgery including the decrease in mean TSH, FT3, and FT4 levels. At 72 h after the surgery, TSH and FT4 were restored to the preoperative levels, while FT3 was partially restored but remained lower than the baseline. The mechanism of early postoperative decrease in TSH and FT3 levels remains unclear in our study with iNPH patients. In cardiac surgery patients, the most suspected mechanism for thyroid dysfunction is a generalized inflammatory response caused due to surgical insult, resulting in organ failure [27,28,29]. In our study, 3 months after the surgery, the percentage of patients with low T3 decreased from 24% to 8%. De Groot reported that reduced FT3 levels may be interpreted as an adaptation to the process of deterioration in patients with chronic disease, with the overall objective of reducing the body’s metabolism [30]. Whether these changes are due to adaptive physiological mechanisms to reduce the metabolic rate during stressful circumstances or a consequence of the underlying process is still a matter of debate [31]. According to the data from the literature, we speculate that VP shunt surgery for iNPH patients has an impact on the adaptation process in chronic disease which in turn may influence the changes in thyroid hormone levels area.

To the best of our knowledge, we could not find any data in the literature regarding the FT3/FT4 ratio in iNPH patients and its association with QoL. According to the data of our study, there were no significant differences between preoperative hormone levels and 3-month follow-up, but the changes in the FT3/FT4 ratio were very clear between the preoperative status and 3-month follow-up period. The FT3/FT4 ratio was the highest in the S4 segment (high FT3/low FT4) preoperatively and it decreased 3 months after the VP shunt surgery. The FT3/FT4 ratio was significantly lower preoperatively in all other segments (S1, S2, S3) and it increased significantly 3 months after the VP shunt surgery. According to our data, the health-related quality of life did not improve after the surgery in iNPH patients who had increased FT3/FT4 ratios (S4 segment, high FT3/low FT4) before the VP shunt surgery. These patients did not have any other major comorbidities (cardiac pathology, etc.), which could influence their quality of life. It has been previously postulated that although intracranial pressure is not chronically elevated in NPH patients, intermittent bursts of intracranial hypertension could contribute to pituitary dysfunction [11,12]. Additionally, Moin et al. discuss that the hypothalamus sustains damage due to local edema, which is associated with altered blood flow and impaired oxygen supply. We speculate that implantation of ventriculoperitoneal shunt lowers the intracranial pressure and the changes in pressure could have an impact on blood flow and oxygen supply in the hypothalamus. Hereby, the ventriculoperitoneal shunt may affect the hypothalamic-pituitary axis, followed by effects on the function of the thyroid gland. The data of our study shows that after the VP shunt surgery the FT3/FT4 ratio in S1, S2, and S3 segments started increasing and approaching the normal values according to age [32].

The mechanism of changes in the FT3/FT4 ratio in the S4 segment remains unclear. It was increased preoperatively when compared with the rest of the segments and the hormone ratio was decreased after the VP shunt surgery. The QoL did not improve in S4 segment patients. Chiarvalloti et al. in their study showed that Alzheimer’s disease patients displayed an abnormal biofeedback regulation in the hypothalamic-pituitary-thyroid axis [33]. They found that Alzheimer’s disease patients showed a trend towards reduced TSH and FT4 levels and loss of correlation between TSH and FT3 or FT4 compared to controls. The study by Choi et al. found an association between lower serum FT4 and cerebral amyloid beta protein deposits in Alzheimer’s disease patients [34]. According to the study by Junkarri et al., the absence of amyloid beta in the frontal cortical biopsy predicted favorable QoL outcomes in iNPH patients after the VP shunt surgery [35]. Considering the data with Alzheimer’s disease patients, we hypothesize that S4 segment patients in our study could have a more complex neurodegenerative disease, although we did not see any apparent clinical symptoms of Alzheimer’s disease either preoperatively or during the follow-up period. Based on the data of the present study, we speculate that patients who had low-normal levels of FT4 and high levels of FT3 preoperatively were at increased risk of non-improvement of their quality of life after the VP shunt surgery compared to those who had high-normal levels of FT4. These findings may help the clinicians to estimate what patients would not benefit from VP shunt surgery.

Measurement of quality-of-life outcomes has become common in surgical research. Overall, QoL improved in 60% of iNPH patients in our study after the VP shunt surgery. These findings are consistent with the literature data from other studies. Junkkari and coworkers reported that at 3-month follow-up 56% of iNPH patients had experienced an improvement in QoL after the VP shunt surgery. At 1 year, only 43% of patients had experienced an improvement in QoL [35]. In our study, the change of QoL has been significantly linked with physical and social domains during the follow-up period of 3 months. According to the results of our study, patients with initially low FT3 had better general health before the operation and low T3 level was not a predictor of poorer QoL in iNPH patients treated with VP shunt surgery.

The present study is limited by the small sample size and relatively short follow-up period. Additionally, we did not investigate other anterior pituitary hormones and did not perform a frontal cortical biopsy for amyloid beta protein. Moreover, this was a single-center study and the findings need to be verified by a larger prospective trial.

## 5. Conclusions

We propose that VP shunt surgery in iNPH patients may have beneficial effects on thyroid hormones. Low FT3 level was not associated with poorer QoL in iNPH patients treated with VP shunt surgery. Patients who had low-normal levels of FT4 and high levels of FT3 preoperatively were at increased risk for non-improvement of their QoL after the VP shunt surgery as compared to patients with high-normal levels of FT4.

## Figures and Tables

**Figure 1 jcm-11-04438-f001:**
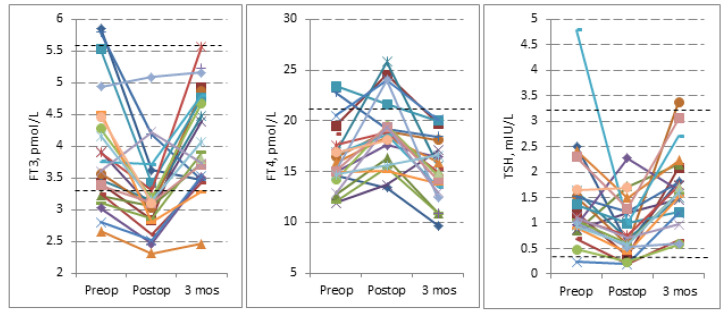
Serum FT3, FT4, and TSH concentrations in iNPH patients from admission until 3-month follow-up after VP shunt surgery. Lines connect hormone determinations of the same subject.

**Figure 2 jcm-11-04438-f002:**
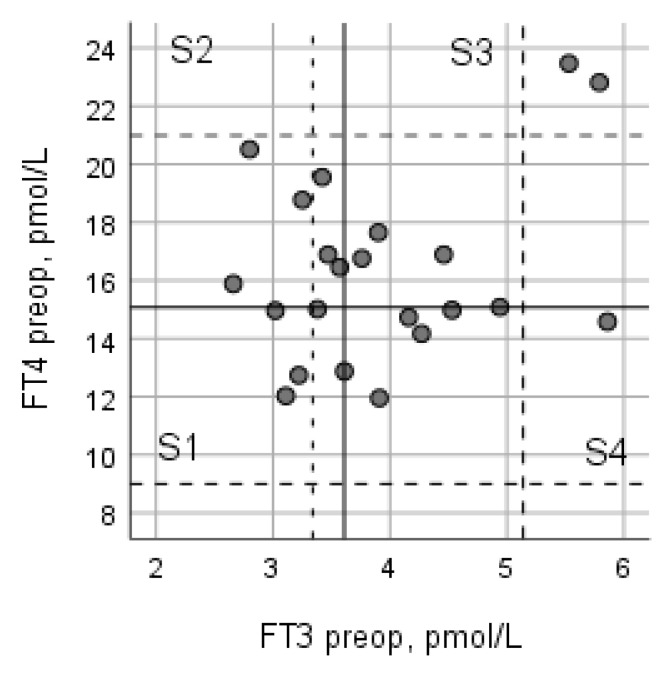
The scatter plot showing the relationship between FT3 and FT4 serum concentrations in iNPH patients before VP shunt surgery. The solid lines show medians that divide the scatter plot into four segments (S1–S4); the dotted lines show the reference values.

**Figure 3 jcm-11-04438-f003:**
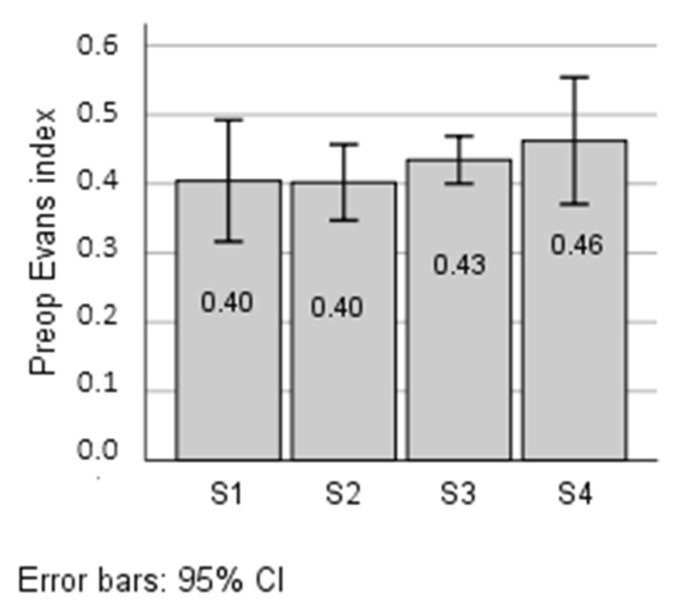
Preoperative Evans index according to the segments of the relationship between FT3 and FT4 levels. FT3: free triiodothyronine; FT4: free thyroxine; S1: LowFT3/LowFT4; S2: LowFT3/HighFT4; S3: HighFT3/HighFT4; S4: HighFT3/LowFT4. Low: below median; high: above median.

**Figure 4 jcm-11-04438-f004:**
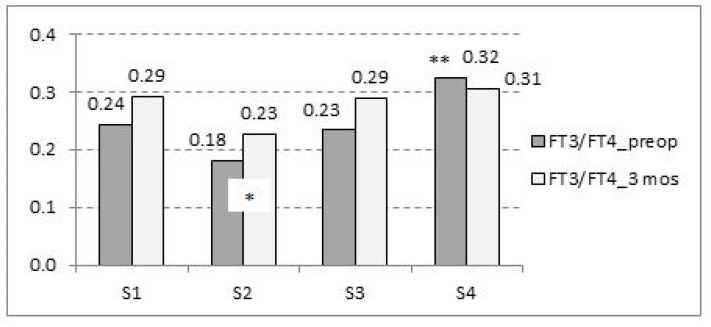
Preoperative FT3/FT4 ratio compared with 3-month follow-up according to the segments of the relationship between FT3 and FT4. * *p* = 0.017 preop vs. 3 months; ** *p* < 0.01 S4 vs. S1, S2, and S3; S1: LowFT3/LowFT4; S2: LowFT3/HighFT4; S3: HighFT3/HighFT4; S4: HighFT3/LowFT4; Low: below median; high: above median.

**Figure 5 jcm-11-04438-f005:**
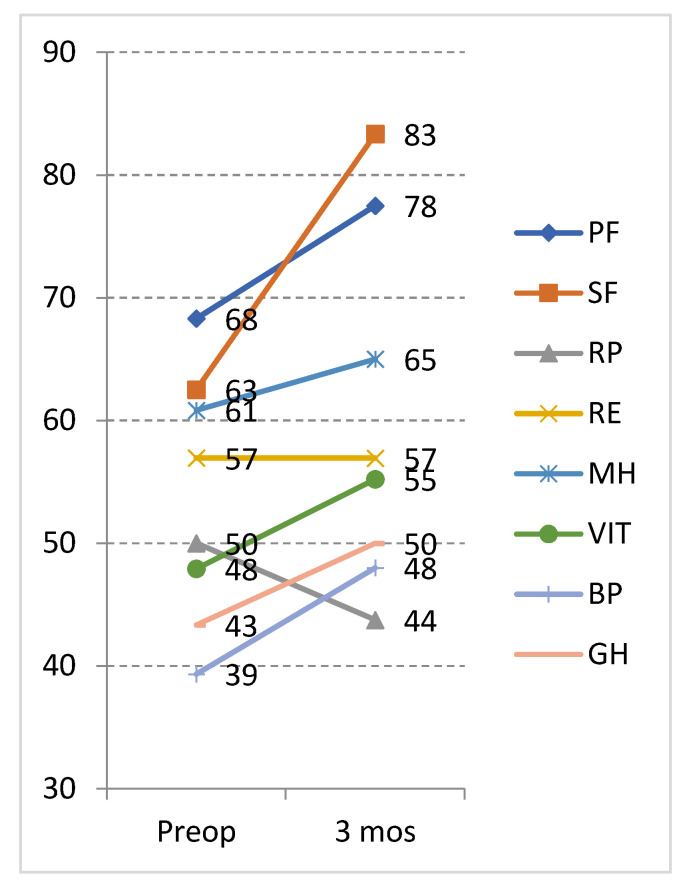
Unadjusted scores of the SF-36 subscales at the baseline and 3 months after VP shunt surgery among patients with low T3 syndrome (N = 6). PF: physical functioning; SF: social functioning, RP: role physical; RE: role emotional, MH: mental health, V: vitality, BP: bodily pain, GH: general health.

**Figure 6 jcm-11-04438-f006:**
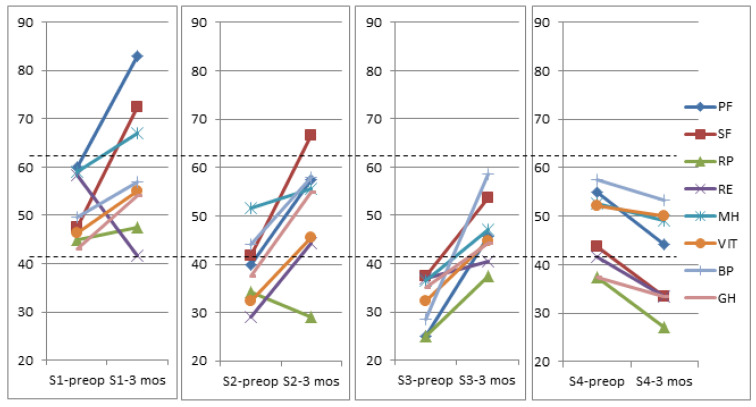
Unadjusted scores of the SF-36 subscales at baseline and 3 months after the VP shunt surgery according to segments of the relationship between preoperative FT3 and FT4 (N = 25). S1: LowFT3/LowFT4; S2: LowFT3/HighFT4; S3: HighFT3/HighFT4; S4: HighFT3/LowFT4; low: below median; high: above median; PF: physical functioning; SF: social functioning; RP: role physical; RE: role emotional; MH: mental health; V: vitality; BP: bodily pain; GH: general health.

**Table 1 jcm-11-04438-t001:** The results of clinical examinations before and 3 months after VP shunt surgery (N = 25).

	Preop ^a^	3 Months ^a^	z	Sig ^b^	Decrease of 1 or More Points
iNPH grading scale:					
*Gait*	2 (2–2.5)	1 (0–1)	−4.49	<0.001	23 (96%)
*Cognition*	2 (1–2)	1 (0–1)	−3.94	<0.001	17 (68%)
*Urinary function*	2 (1–3)	1 (1–2)	−3.05	0.002	10 (40%)
*Total*	5 (4–7.5)	3 (2–4.5)	−4.41	<0.001	
TUG, secs (N = 24)	20 (18.25–23.5)	15 (13–18.5)	−4.31	<0.001	
MMSE, score	23 (21.5–24)	25 (24–28)	−4.39	<0.001	

^a^: median (25% and 75% IQR). ^b^ Wilcoxon signed-rank test. TUG: the timed up-and-go test; MMSE: Mini-Mental State Examination.

**Table 2 jcm-11-04438-t002:** Evans index and thyroid hormone levels measured pre- and postoperatively, and 3 months after VP shunt surgery.

	Preop	Postop	3 Months	*p*-Value		
	1	2	3	1 vs. 2	2 vs. 3	1 vs. 3
Evans index	0.427 ± 0.063	0.411 ± 0.067	0.402 ± 0.007	0.002	0.028	<0.001
*Thyroid hormones*						
FT3, pmol/L	3.93 ± 0.92	3.21 ± 0.64	4.09 ± 0.76	<0.001	<0.001	0.454
Median (range)	3.68 (2.66–5.86)	3.11 (2.30–5.10)	3.90 (2.47–5.58)			
FT4, pmol/L	16.18 ± 3.18	18.93 ± 3.50	15.29 ± 3.01	0.001	<0.001	0.077
Median (range)	15.40 (11.95–23.47)	18.82 (13.39–25.83)	14.93 (9.66–20.08)			
TSH, mIU/L	1.41 ± 0.95	0.87 ± 0.55	1.69 ± 0.74	0.019	<0.001	0.119
Median (range)	1.10 (0.24–4.79)	0.69 (0.19–2.27)	1.59 (0.58–3.38)			
Low T3 syndrome ^a^ (*n* = 6)						
Low T3, *n* (%)	6 (24)	16 (64)	2 (8)	0.002 ^b^	<0.001 ^b^	0.219 ^b^
Low T3, pmol/L	3.01 ± 0.23	2.63 ± 0.27	3.43 ± 0.50	0.004	0.002	0.038

Values are presented as mean ± standard deviation or number (%). Reference intervals: FT3: 3.34–5.14 pmol/L; FT4: 9–21.07 pmol/L; TSH (thyroid-stimulating hormone): 0.4–3.6 mU/L. ^a^ Low T3 syndrome when FT3 ≤ 3.34 pmol/L; Cochran’s Q test < 0.001; ^b^ paired McNemar’s test.

**Table 3 jcm-11-04438-t003:** SF-36 subscale scores at baseline and 3 months after VP shunt and effect size (d).

					Change (3 Months–Preop)	
SF-36 Subscale	Preop	3 Months	Wilcoxon-Z	*p*	Mean, Score	Effect Size
Physical functioning	43.54 ± 30.52	56.04 ± 29.70	−2.19	0.028	12.50	0.4096
Social functioning	42.18 ± 27.28	55.72 ± 25.53	−2.45	0.014	13.54	0.4962
Role physical	34.63 ± 23.31	34.89 ± 25.19	−0.74	0.454	0.26	0.0112
Role emotional	40.62 ± 30.02	39.93 ± 28.96	−0.24	0.808	−0.69	−0.0231
Mental health	48.95 ± 21.05	53.87 ± 16.24	−2.09	0.036	4.91	0.2335
Vitality	40.10 ± 21.24	48.38 ± 19.24	−2.52	0.012	8.28	0.3903
Bodily pain	44.00 ± 22.35	56.79 ± 23.28	−2.31	0.021	12.79	0.5722
General health	37.91 ± 15.52	46.25 ± 14.68	−2.32	0.020	8.33	0.5367

Data represent mean ± standard deviation. Effect size, Cohen’s d.

**Table 4 jcm-11-04438-t004:** The effect size of changes in SF-36 subscales scores before and 3 months after the VP shunt surgery according to segments of the relationship between preoperative FT3 and FT4.

	S1 Low FT3/Low FT4	S2 Low FT3/High FT4	S3 High FT3/High FT4	S4 High FT3/Low FT4
FT3/FT4 Preop Mean (95% CI)	0.243 (0.205–0.281)	0.179 (0.148–0.209)	0.233 (0.2166–0.263)	0.322 (0.279–0.367) *
Physical functioning	0.753	0.573	0.753	−0.35
Social functioning	0.916	0.916	0.916	−0.39
Role physical	0.107	−0.223	0.107	−0.45
Role emotional	−0.555	0.508	−0.555	−0.27
Mental health	0.379	0.182	0.379	−0.16
Vitality	0.412	0.628	0.412	−0.09
Bodily pain	0.331	0.626	0.331	−0.18
General health	0.708	1.127	0.708	−0.26

It has been suggested that approximately 0.5 of a standard deviation of the scale score is a reasonable statistical cut point for determining significant change. The interpretation of effect size Cohen’s d: (none, very small) (0.20>); small 0.20≤; medium 0.50≤; large 0.80≤; low: below median; high: above median; median FT3 = 3.68 pmol/L; median FT4 = 15.4 pmol/L. * all *p* < 0.01, in comparison FT3/FT4 ratio in S4 with S1, S2, and S3.

**Table 5 jcm-11-04438-t005:** Results of multiple regression analysis to predict postoperative 3-month follow-up outcomes.

Dependent Variable ^a^	Independent Variable	β	t	*p*	R^2^	Model
PF	PF preop	0.544	2.429	0.029	0.246	<0.001
SF	SF preop	0.665	4.415	<0.001	0.542	<0.001
	FT3/FT4 preop	–0.531	–3.52	0.002		
MH	MH preop	0.882	9.27	<0.001	0.810	<0.001
	Age, year	–0.229	–2.41	0.026		
V	V preop	0.648	5.397	<0.001	0.756	<0.001
	Age	–0.404	–3.123	0.006		
	Gender (M0, W1)	0.269	2.181	0.043		
BP	BP preop	0.417	2.049	0.054	0.132	0.138
GH	FT3 preop	–0.458	–2.302	0.032	0.17	0.032

Adjusted for: age, gender, preoperative FT3, FT4, FT3/FT4 ratio, Evans index, and preoperative score of the SF-36 subscale. The rating scale was scaled from 0 to 100. ^a^ SF-36 subscale scores 3-months after the shunting operation. PF: physical functioning; SF: social functioning, MH: mental health, V: vitality, BP: bodily pain, GH: general health.

## Data Availability

Not applicable. All the data supporting our findings have been presented in the tables in this article. This is a series of studies, and the study of this population is underway, so this part of the original data is not publicly disclosed at the present stage.

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
