# Peer review of "Thyroid Hormones and Health-Related Quality of Life in Normal Pressure Hydrocephalus Patients before and after the Ventriculoperitoneal Shunt Surgery: A Longitudinal Study"

_jcm, 2022, doi:10.3390/jcm11154438_

Round 1

Reviewer 1 Report

Interesting study of 25 patients who underwent VPS. TSH, FT4, FT3 were measured before, after, and 3 months after shunt surgery. The authors found a upright U-curve for TSH and FT3 across the 3 time points, vs. an inverted U-curve for FT4. In general, there were no statistically significant differences between aggregate baseline and 3 month thyroid hormone values, except for patients with low initial T3. When stratified by low/high baseline FT3 and FT4, general health, physical and social functioning effect sizes were improved for all except the high FT3/low FT4 group.

The findings are interesting and potentially impactful in spurring larger studies with better control groups. Do the authors propose routine analysis of thyroid hormones as part of the NPH workup?

Line 290 states "RF" however i cannot find the explanation of this abbreviation anywhere in the text. 

Author Response

Thank you very much for your kind words about our paper. We are grateful for the time and energy you expended on our behalf. Any revisions made to the manuscript were marked up using the “Track Changes” function.

According to the data of our study, patients who had low-normal levels of FT4 and high levels of FT3 preoperatively were at increased risk for non-improvement of their QoL after the VP shunt surgery as compared to the patients with high-normal levels of FT4. Although the present study is limited by the small sample size, the findings of the study are very clear. We think that routine analysis of thyroid hormones could be a part of NPH workup.

„RF“ was changed to „RP“ (Role physical) (line 291).

The revised manuscript underwent the professional English language editing.

Reviewer 2 Report

The article is a rare but interesting study. The authors take an endocrine axis perspective and look at the relationship between changes in thyroid related hormones and quality of life in NPH patients after VP.

The discussion in the article talks about other studies that have shown no statistical difference between preoperative and postoperative FT3 and FT4. The possible reasons for the difference in the authors' results in this case are worth discussing.

In addition the pathophysiological mechanisms behind the trend of thyroid related hormone changes could be properly discussed, could such changes provide new insights and directions to the pathogenesis of iNPH? Or is there any connection with the existing pathogenesis of iNPH (e.g. glial lymphatic system, neuroinflammation, impaired perfusion of periventricular tissue, etc.).

Furthermore, the surgery is an invasive operation, and does this operation itself lead to changes in thyroid hormones? And are the changes in the patient's SF-36 associated with symptomatic improvement after treatment (not limited to ventriculoperitoneal shunt surgery)? This point was not reflected in the article.

Author Response

Thank you very much for your kind words about our paper. We are grateful for the time and energy you expended on our behalf. We have studied the comments carefully and have made corrections which we hope will meet with the approval. Any revisions made to the manuscript were marked up using the “Track Changes” function.

The discussion in the article talks about other studies that have shown no statistical difference between preoperative and postoperative FT3 and FT4. The possible reasons for the difference in the authors' results in this case are worth discussing.

ANSWER. According to the Ucler data, there were no significant differences between preoperative and postoperative FT3 and FT4 values in the patient group. But that study did not evaluated the thyroid hormones in the early postoperative period. Therefore the following sentence was clarified in the DISCUSSION section (line 371-374): “Ucler and coworkers presented the study with hydrocephalic newborns who underwent a VP shunt surgery and thyroid hormones were investigated before and 30 and 90 days after the surgery.” Following sentence was included into the DISCUSSION section (line 377-379): „Importantly, Moin’s study found that endocrine dysfunction may be reversible in some NPH patients 3 months after the neurosurgical correction.“ The thyroid hormones were evaluated preoperatively, the next day after the operation and 3 months after the operation in our study. The significant changes occurred the next day after the surgery, including a decrease in TSH, FT3 and an increase in FT4. In general, there were no statistically significant differences between the baseline and 3 months thyroid hormone values, except for the patients with low initial T3.

In addition the pathophysiological mechanisms behind the trend of thyroid related hormone changes could be properly discussed, could such changes provide new insights and directions to the pathogenesis of iNPH? Or is there any connection with the existing pathogenesis of iNPH (e.g. glial lymphatic system, neuroinflammation, impaired perfusion of periventricular tissue, etc.).

ANSWER. The etiology of iNPH is not clear and pathophysiology remains vague, too. iNPH combines several pathogenetic factors leading to a self-reinforcing vicious circle. Therefore we decided not to focus on pathophysiology so much in this study. Following sentence was included into the DISCUSSION section (line 425-429): „It has been previously postulated that although intracranial pressure is not chronically elevated in NPH patients, intermittent bursts of intracranial hypertension could contribute to pituitary dysfunction“

Furthermore, the surgery is an invasive operation, and does this operation itself lead to changes in thyroid hormones?

ANSWER. According to the literature, in cardiac surgery patients the most suspected mechanism for thyroid dysfunction is generalized inflammatory response caused due to surgical insult, resulting in organ failure. It was found that the most significant changes occurred 24 hours after the surgery including the decrease in mean TSH, FT3 and FT4 levels.  The data with hydrocephalus patients is very limited because the existing studies do not focus on early postoperative changes in thyroid hormones. From ours experience, the changes in thyroid hormones according to the treatment method for hydrocephalus  (ventriculoperitoneal shunt vs endoscopic third ventriculostomy) were different preoperatively and the next day after the operation (part of the data was presented in the 17th European Congress of Neurosurgery, Venice, 2017. M. Urbonas et al. “Changes of thyroid hormones after ventriculoperitoneal shunting and endoscopic third ventriculostomy”, EP684).

And are the changes in the patient's SF-36 associated with symptomatic improvement after treatment (not limited to ventriculoperitoneal shunt surgery)? This point was not reflected in the article.

ANSWER. This point was not the aim of this study, therefore it was not reflected in the article. However, in some cases we noticed improvement in iNPH scale although patient’s self-reported health-related quality of life did not change or worsened after the treatment.

Thank you very much for taking the time and energy to help us to improve the paper. We hope that our supplementary analysis and revised focus will help you to improve your opinion of our work.

Reviewer 3 Report

This study is well designed and interesting., although the interpretation of the results is a little bit complicated. And if possible, the authors would be pleased to have more sample size to lead precise conclusion of the effect of thyroid axis on hydrocephalus. The indication of VP shunt was well described in the Method section and the results are also well presented. 

Author Response

Thank you very much for your kind words about our paper. We are grateful for the time and energy you expended on our behalf.

Any revisions made to the manuscript were marked up using the “Track Changes” function.

Reviewer 4 Report

-

      The 2 sentences contained in lines 100-103 should be moved to the following 2.2 paragraph about SF-36 questionnaire

Author Response

Thank you very much for your kind words about our paper. We are grateful for the time and energy you expended on our behalf. Any revisions made to the manuscript were marked up using the “Track Changes” function.

The 2 sentences contained in lines 100-103 should be moved to the following 2.2 paragraph about SF-36 questionnaire

ANSWER. The sentences „All patients within 3 days before and 3 months after the surgery were evaluated for quality of life with SF-36 questionnaire. Patients were asked to complete the SF-36 questionnaire themselves and were given an opportunity to ask questions if any“ were moved to the following 2.2 paragraph (lines 105 – 107).